# Biallelic structural variations within *FGF12* detected by long-read sequencing in epilepsy

Sachiko Ohori[1,2], Akihiko Miyauchi[3], Hitoshi Osaka[3], Charles Marques Lourenco[4,5], Naohiro Arakaki[6,7], Toru Sengoku[8], Kazuhiro Ogata[8], Rachel Sayuri Honjo[9], Chong Ae Kim[9], Satomi Mitsuhashi[10], Martin C Frith[11,12,13], Rie Seyama[1,14], Naomi Tsuchida[1,15], Yuri Uchiyama[1,15], Eriko Koshimizu[1], Kohei Hamanaka[1], Kazuharu Misawa[1], Satoko Miyatake[1,16], Takeshi Mizuguchi[1], Kuniaki Saito[6,7], Atsushi Fujita[1], Naomichi Matsumoto[1]

We discovered biallelic intragenic structural variations (SVs) in *FGF12* by applying long-read whole genome sequencing to an exome-negative patient with developmental and epileptic encephalopathy (DEE). We also found another DEE patient carrying a biallelic (homozygous) single-nucleotide variant (SNV) in *FGF12* that was detected by exome sequencing. *FGF12* heterozygous recurrent missense variants with gain-of-function or heterozygous entire duplication of *FGF12* are known causes of epilepsy, but biallelic SNVs/SVs have never been described. *FGF12* encodes intracellular proteins interacting with the C-terminal domain of the alpha subunit of voltage-gated sodium channels 1.2, 1.5, and 1.6, promoting excitability by delaying fast inactivation of the channels. To validate the molecular pathomechanisms of these biallelic *FGF12* SVs/SNV, highly sensitive gene expression analyses using lymphoblastoid cells from the patient with biallelic SVs, structural considerations, and *Drosophila* in vivo functional analysis of the SNV were performed, confirming loss-of-function. Our study highlights the importance of small SVs in Mendelian disorders, which may be overlooked by exome sequencing but can be detected efficiently by long-read whole genome sequencing, providing new insights into the pathomechanisms of human diseases.

## Introduction

Exome sequencing (ES) has significantly contributed to the understanding of genetic causes in Mendelian disorders (Bamshad et al, 2011; Tennessen et al, 2012). However, ES has been less powerful in detecting structural variations (SVs) that involve intronic, GC-rich or repetitive regions than detecting single-nucleotide variants (SNVs) (Zhao et al, 2021; Marwaha et al, 2022). In contrast, long-read sequencing (LRS) enables us to obtain reads spanning entire or partial SVs throughout the entire genome, with the advantage of direct characterization of complex SVs, overcoming the difficulties of SV detection by ES (Miao et al, 2018; Mizuguchi et al, 2019; Lei et al, 2020; Marwaha et al, 2022). For SV analysis, targeted-LRS is cost-effective, but this methodology is applicable solely in instances where the genomic region of interest has been predetermined (Miller et al, 2021). Therefore, long-read "whole genome" sequencing (LRWGS) should be applied to unsolved patients with no suspected candidate loci. To efficiently detect pathogenic SVs using LRWGS, we previously established the analysis pipeline, dnarrange, with which we can detect patient-specific SVs by eliminating polymorphic SVs found in control datasets (Mitsuhashi et al, 2020). Then, we demonstrated that dnarrange can identify pathogenic SVs in a genome-wide and nontargeted manner (Ohori et al, 2021).

Fibroblast growth factor 12, encoded by *FGF12*, is a member of the fibroblast growth factor (FGF) homologous factor family (such as FGF11, FGF12, FGF13, and FGF14). It is an intracellular protein and interacts with the C-terminal domain of the alpha subunit of voltage-gated sodium channels (Na$_v$s) 1.2, 1.5, and 1.6 to promote excitability by delaying fast inactivation of the channels (Goldfarb, 2012; Wildburger et al, 2015; Pablo & Pitt, 2016).

A recurrent de novo heterozygous missense variant, [c.155G>A p.(Arg52His)] of *FGF12* has been reported in individuals with developmental and epileptic encephalopathy (DEE) and cerebellar

[1]Department of Human Genetics, Yokohama City University Graduate School of Medicine, Yokohama, Japan   [2]Department of Genetics, Kitasato University Hospital, Sagamihara, Japan   [3]Department of Pediatrics, Jichi Medical School, Shimotsuke, Japan   [4]Neurogenetics Department, Faculdade de Medicina de São José do Rio Preto, São Jose do Rio Preto, Brazil   [5]Personalized Medicine Department, Special Education Sector at DLE/Grupo Pardini, Belo Horizonte, Brazil   [6]Department of Chromosome Science, National Institute of Genetics, Research Organization of Information and Systems (ROIS), Shizuoka, Japan   [7]Graduate Institute for Advanced Studies, SOKENDAI, Shizuoka, Japan   [8]Department of Biochemistry, Yokohama City University Graduate School of Medicine, Yokohama, Japan   [9]Unidade de Genética Médica do Instituto da Criança, Hospital das Clinicas HCFMUSP, Faculdade de Medicina, Universidade de Sao Paulo, Sao Paulo, Brazil   [10]Department of Neurology, St. Marianna University School of Medicine, Kawasaki, Japan   [11]Artificial Intelligence Research Center, National Institute of Advanced Industrial Science and Technology (AIST), Tokyo, Japan   [12]Graduate School of Frontier Sciences, University of Tokyo, Kashiwa, Japan   [13]Computational Bio Big-Data Open Innovation Laboratory, AIST, Tokyo, Japan   [14]Department of Obstetrics and Gynecology, Juntendo University, Tokyo, Japan   [15]Department of Rare Disease Genomics, Yokohama City University Hospital, Yokohama, Japan   [16]Department of Clinical Genetics, Yokohama City University Hospital, Yokohama, Japan

Correspondence: naomat@yokohama-cu.ac.jp

atrophy (#617166; MIM) (Siekierska et al, 2016; Takeguchi et al, 2018). Functional analyses of *FGF12* have shown that the gain-of-function (GOF) effect of the missense variant on Na$_V$ causes epilepsy through enhanced modulation of channel inactivation gating, increasing neuronal excitability (Siekierska et al, 2016). In addition, entire *FGF12* duplication has been suggested to cause severe epilepsy, implying a pathomechanism similar to the GOF effect of *FGF12* (Oda et al, 2019).

In this study, we applied LRWGS and dnarrange to an ES-negative DEE patient (patient 1) and discovered biallelic intragenic SVs in *FGF12*. We then used highly sensitive droplet digital polymerase chain reaction (ddPCR) to observe diminished *FGF12* expression caused by the SVs. This technique overcomes the low level of *FGF12* expression in lymphoblastoid cell lines (LCLs), which is far below the sensitivity of regular real-time PCR (Heredia et al, 2013; Taylor et al, 2017), and enabled us to confirm altered allele-specific expression (ASE). In addition, we found another DEE patient (patient 2) carrying a homozygous (biallelic) missense variant in *FGF12* by ES. The nature of the missense variant was evaluated by structural considerations and *Drosophila* in vivo functional studies.

## Results

### Clinical features

Patient 1 is an 8-yr-10-mo-old girl, the third child to non-consanguineous Japanese healthy parents in family 1 (Fig 1A). She was delivered at term after an uneventful pregnancy (birth-weight, 3,222 g). She showed infantile spasms at the age of 5 mo and absence of head control. Electroencephalography (EEG) exhibited hypsarrhythmia (occurring once every 10 s), which was consistent with West syndrome. She was treated orally with vitamin B6 (30 mg/kg/d). As there was no improvement in symptoms and she was gradually showing regression, she was admitted to the hospital at 7 mo for adrenocorticotropic hormone therapy (0.0125 mg/kg/d). Although hypsarrhythmia disappeared from the EEG after a 3-wk therapy, her spasms persisted. Then, valproic acid (VPA) (30 mg/kg/d) and zonisamide (ZNS) (4 mg/kg/d) therapies were initiated. The brain magnetic resonance imaging (MRI) was almost normal at 7 mo (Fig 2A–E), but they newly identified a left subdural hematoma at 8 mo, which was considered a hemorrhage subsequent to cerebral atrophy. She exhibited focal motor seizures followed by bilateral tonic–clonic and status epilepticus. Phenytoin (PHT) (7.5 mg/kg/d) was partially effective for generalized status epilepticus and the tonic–clonic seizures with desaturations disappeared. Intravenous midazolam (MDZ) therapy (0.1 mg/kg) was also needed when status epilepticus occurred. At 10 mo, a brain MRI showed a right subdural hematoma. As seizures reoccurred following the reduction of MDZ therapy, high-dose phenobarbital (PB) therapy (8 mg/kg/d) was started, which improved the symptoms. Levetiracetam (10 mg/kg/d) and clobazam (0.3 mg/kg/d) were invalid.

Since then, her seizures have been controlled with VPA (45 mg/kg/d), ZNS (7.5 mg/kg/d), PHT (7.5 mg/kg/d), and PB (4 mg/kg/d), with clusters of convulsions only in a febrile condition. At the age of 1 yr 7 mo, her brain MRI showed severe cerebral atrophy with no

obvious cerebellar atrophy (Fig 2F–J) and epileptic encephalopathy was diagnosed. Thereafter, her seizures got more frequent during febrile episodes, but improved with fever amelioration and she was discharged at 2 yr of age. On the last visit at 5 yr 6 mo, her seizures were well controlled, but severe developmental delay and intellectual disability (ID) remained (Table 1). Conventional karyotyping or chromosomal microarray (CMA) has not been performed.

Patient 2 was a boy, the third child to healthy consanguineous Portuguese and African parents who were third-degree cousins in family 2 (Fig 1B). He was delivered at full term by elective cesarean section (social indication) after an uneventful pregnancy (birth-weight, 3,290 g). He was referred for evaluation of hypotonia and abnormal movements including tremors in his hands and head at the age of 2 mo. His brain computed tomography was normal at that time and physical therapy was started at 4 mo. At 5–6 mo, his spasms and neurodevelopmental stagnation were noted by his parents and EEG suggested West syndrome or multifocal epileptic encephalopathy. VPA therapy (15 mg/kg/d) was started but there was only partial response. His seizures persisted despite increasing the dose of VPA (up to 75 mg/kg/d) and adding topiramate (20 mg/kg/d) and vigabatrin (80 mg/kg/d). After PB treatment (18 mg/kg/d), his seizures were partially controlled with carbamazepine (45 mg/kg/d), although cannabidiol treatment (30 mg/kg/d) did not improve them. Following the age of 2 yr, his hypotonia and tremors gradually changed to spasticity and myoclonic jerks with dysmorphic facial features including a broad forehead and pointed chin (Fig 1C).

Brain MRI at 12 mo and 1 yr 7 mo showed mild cerebral atrophy and normal myelination without cerebellar atrophy (Fig 2K–R), and at 6 yr of age showed gray matter atrophy along with cortical atrophy and secondary peritrigonal white matter involvement (Fig 2S–U). Conventional karyotyping or CMA was not performed. At the age of 11 yr, he died of severe pneumonia caused by COVID-19 (Table 1). Video 1 shows his seizure (epileptic spasm) at the age of 1 yr 2 mo, when he had seizures of this nature with high frequency (~100 per day).

### Genetic analysis of patient 1

Trio-based ES for this patient revealed no pathogenic SNVs or CNVs on examination by XHMM (the eXome Hidden Markov Model) (Fromer et al, 2012; Miyatake et al, 2015). Next, using a nanopore long-read sequencer PromethION, genomic DNA of peripheral blood leukocytes from patient 1 was sequenced. A total of 4,202,503 long reads were obtained with 90,602,996,990 bases (~30× read coverage of Genome Reference Consortium Human Build 38, GRCh38), with mean and median read lengths of 21,559 and 18,902 bases, respectively. We applied dnarrange to find SVs specific to this patient, excluding polymorphic SVs seen in 30 control individuals, and as a result, 65 groups of patient-specific SV reads were extracted. The patient-specific SVs included mostly retro-transposon insertions, deletions, tandem multiplications, and tandem repeat expansions. We focused on 20 SVs involving protein-coding genes (Table S1) and were particularly interested in *ZNF292*- and *FGF12*-related SVs.

Regarding the heterozygous tandem duplication (size: 3.2 kb, chr6: 87,201,507–87,204,665) within intron 1 of *ZNF292* (Fig S1A),

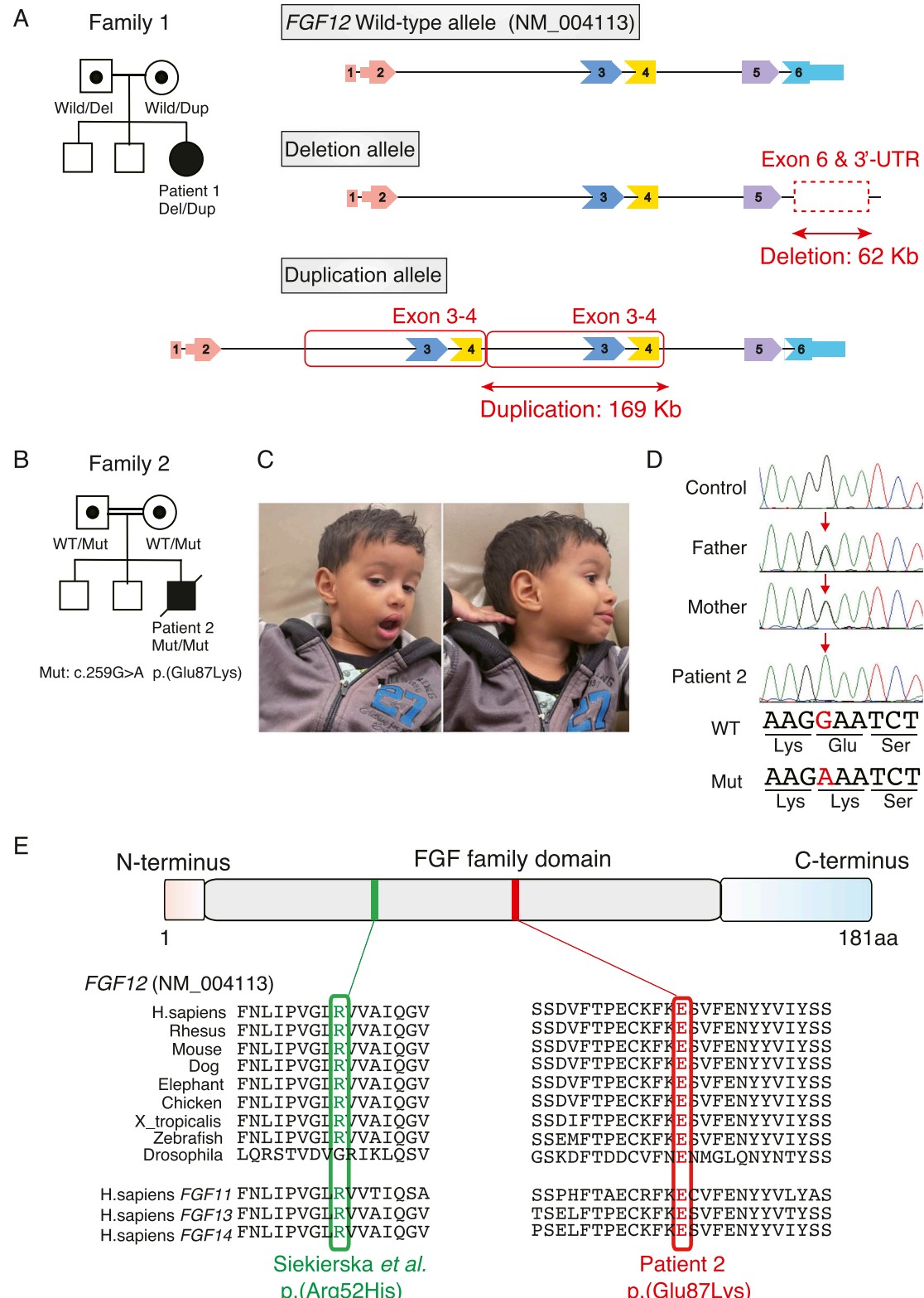

**Figure 1. Characterization of biallelic intragenic SVs (family 1) and a homozygous SNV (family 2) in *FGF12*.**
**(A)** The pedigree and biallelic intragenic SVs in *FGF12* of family 1. Wild-type and aberrant alleles harboring intragenic deletion and duplication in patient 1 and her parents. *FGF12* transcripts consist of two main isoforms, isoform-A (NM_021032) and isoform-B (NM_004113), containing five and six exons, respectively, that differ in the N-terminal sequences. In this study, all *FGF12* variants are described based on the reference sequence NM_004113, because isoform-B is most abundant in the human brain (GTEx portal) and is also selected as a representative transcript by the Matched Annotation from NCBI and EMBL-EBI (MANE) project. Wild, *FGF12* wild-type allele;

breakpoint PCR showed that it was inherited from the healthy father (Fig S1B). Thus, we considered this tandem duplication as likely benign, because although *ZNF292* variants cause autosomal dominant intellectual developmental disorder (# 619188; MIM), the father with this *ZNF292* intronic tandem duplication is healthy. No DNA from the siblings was available to confirm the variant.

Then, we focused on two different SVs in *FGF12*, a 62-kb partial deletion involving the last exon (exon 6) and the 3′ UTR (untranslated region) (chr3:192,083,678–192,145,703) and a 169-kb intragenic-tandem duplication involving exons 3 and 4 (chr3:192,330,351–192,499,579) (Figs 1A, S2A–C, and S3A–C). We also applied optical genome mapping (OGM) to patient 1 and confirmed the presence of biallelic SVs in *FGF12* (Fig 3), although there were some discrepancies in the sizes and breakpoints detected by LRWGS and OGM (Table S2). We further investigated the breakpoints, based on those identified by LRWGS, using breakpoint PCR, Sanger sequencing, and jNord with the exome data (Nord et al, 2011; Uchiyama et al, 2021), confirming that they were each inherited from one of her parents and were thus biallelic SVs (Figs S2D and E, S3D and E, and S4). The paternal 62-kb deletion allele may result in altered expression as the last exon and 3′ UTR are missing. In contrast, the maternal tandem duplication allele may possibly be subjected to nonsense-mediated mRNA decay (NMD), as the duplicated segment containing exons 3 and 4 (a total of 215 bases of the coding sequence, thus out of frame) will lead to a premature termination codon in exon 5 (Fig S3F). Thus, the *FGF12* expression level was presumed to be diminished in patient 1. Furthermore, no homozygous loss-of-function (LOF) variants in *FGF12* were found in gnomAD (Genome Aggregation Database), implying the detrimental effects of biallelic LOF SVs of *FGF12* in humans.

### The expression analysis of *FGF12* mRNA in LCL of patient 1 using ddPCR

*FGF12* transcripts in LCL of patient 1 were significantly decreased (0–0.50 copies/$\mu$l; median, 0 copies/$\mu$l), compared with those of her parents and an unrelated control ($P < 0.01$ in all comparisons, Mann–Whitney $U$ tests) (Fig 4A). We also confirmed the substantial *FGF12* expression in LCLs of controls compared with those of patient 1 (Fig 4C). Therefore, biallelic SVs in patient 1 resulted in complete LOF of *FGF12*.

### ASE (allele-specific expression) analysis of the normal allele in LCL of the father of patient 1 using ddPCR

As *FGF12* mRNA expression in LCL of the father was higher than that of the control (Fig 4A), we conducted ASE analysis by ddPCR. We evaluated the expression of the normal allele in the father using TaqMan probes (one of which was within the deletion), showing a similar level

(0.59–1.8 copies/$\mu$l; median, 0.93 copies/$\mu$l) compared with the controls (Fig 4E). This suggested the possible transcriptional complementation of the normal allele in the father up to a similar level as the biallelic normal allele expression in controls. The TaqMan probe used in this assay spanned exon 5 and the last exon (exon 6), the latter of which was involved in the deletion allele, therefore eliminating aberrant transcripts of the deletion allele (Fig 4F and Table S3).

### Expression analysis of the abnormal allele harboring a tandem duplication in LCL of the mother of patient 1

The normal 213-bp PCR product of complementary DNA (cDNA) in *FGF12* was detected in the control, patient 1, and her mother (Fig 5A and B). The normal transcript in patient 1 was nearly unseen (Fig 5B), demonstrating that the *FGF12* mRNA expression level was extremely low. In contrast, the aberrant 428-bp PCR product harboring the tandem duplication was seen in both patient 1 and her mother (Fig 5A and B).

After treatment of LCL with cycloheximide (CHX), the electropherogram signal peaks by Sanger sequencing of the aberrant transcript were higher than those with no CHX treatment (Fig 5C). This suggests that the abnormal transcript is subjected to NMD.

### Genetic analysis of patient 2

Trio-based ES identified a homozygous variant [c.259G>A p.(Glu87Lys)] in *FGF12* but no other abnormalities, which was subsequently confirmed by Sanger sequencing (Fig 1D). This variant was predicted to be damaging (score = 0.000) in SIFT, probably damaging (score = 1.0) in PolyPhen-2, and scored 32 in CADD. The affected amino acid residue, Glu87, is highly evolutionarily conserved among species (Fig 1E) and the variant was not registered in gnomAD. Taken together, the variant in *FGF12* was considered a variant of unknown significance (PM2, PM3, PP3) based on the ACMG guidelines (Richards et al, 2015).

### Structural consideration of FGF12-Na$_V$1.5/FGF13-Na$_V$1.2

To gain structural insight into the *FGF12* Glu87Lys variant, we analyzed the reported structure of the FGF12–Na$_V$1.5–calmodulin complex (Fig 6A and C, PDB ID 4JQ0) (Wang et al, 2014). Since some side chains are not visible in this structure because of the relatively low resolution (3.84 Å), we also analyzed a homologous structure containing FGF13, Na$_V$1.2, and calmodulin determined at 3.02 Å resolution (Fig 6B and D, PDB ID 4JPZ) (Wang et al, 2012). In these structures, FGF12 Glu87 and its equivalent residue (FGF13 Glu92) are located at the interface with the Na$_V$ proteins and form

---

Del, *FGF12* intragenic deletion allele; Dup, *FGF12* intragenic tandem duplication allele. **(B)** The pedigree in family 2. WT, wild-type allele; Mut, variant allele harboring c.259G>A p.(Glu87Lys). **(C)** Facial photographs of patient 2 at the age of 4 yr, showing a broad forehead and pointed chin. **(D)** Electropherograms of the homozygous missense variant in *FGF12* of family 2. Patient 2 has a homozygous missense variant, c.259G>A p.(Glu87Lys), inherited from each of his parents. The corresponding nucleotides and amino acid residues are shown under the electropherograms. WT, wild-type allele; Mut, variant allele harboring c.259G>A p.(Glu87Lys); Red arrows, location of the variant. **(E)** Schematic presentation of the FGF12 protein corresponding to isoform-B (NM_004113) with functional domains showing a homozygous SNV in patient 2 together with a recurrent gain-of-function variant. Red and green variants represent p.Glu87Lys in patient 2 and a recurrent gain-of-function variant, p.Arg52His, respectively. Both pathogenic variants occur at amino acids that are evolutionarily well conserved among species (Vertebrate Multiz Alignment & Conservation [100 Species] and Multiz Alignment & Conservation [124 insects] on UCSC genome browser). aa, amino acids.

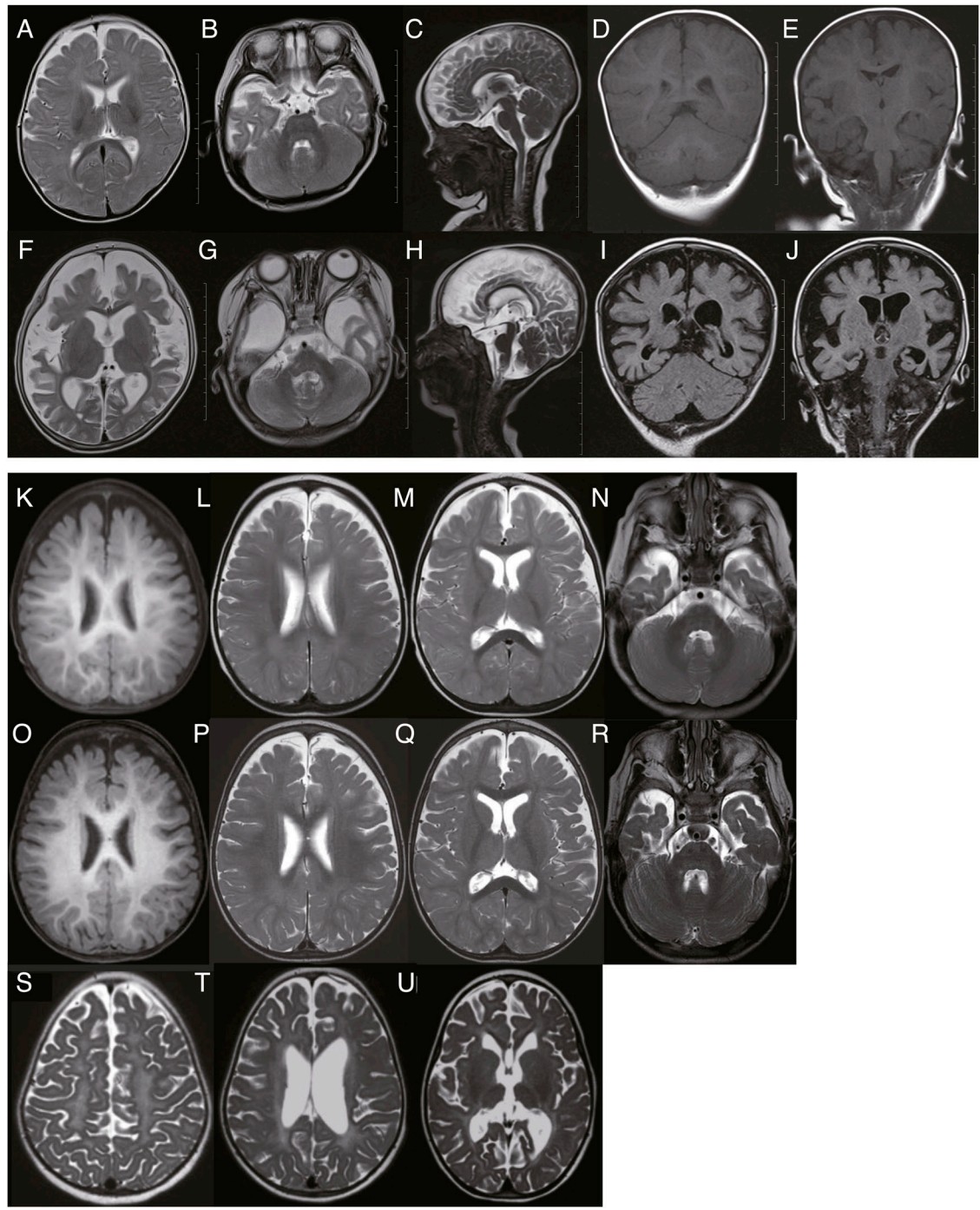

**Figure 2. Brain MRI images of patients 1 and 2.**
**(A, B, C, D, E)** MRI at the age of 7 mo was mostly normal in patient 1. 1 scale: 1 cm. **(F, G, H, I, J)** MRI at 1 yr and 7 mo indicated severe diffuse cerebral atrophy without cerebellar atrophy in patient 1.1 scale: 1 cm. **(K, L, M, N)** MRI at the age of 12 mo indicated mild cerebral atrophy and normal myelination without cerebellar atrophy in patient 2. **(O, P, Q, R)** MRI at 1 yr and 7 mo indicated mild cerebral atrophy and normal progression of the myelination without cerebellar atrophy in patient 2. **(S, T, U)** MRI at 6 yr indicated gray matter atrophy along with cortical atrophy and secondary peritrigonal white matter involvement in patient 2. Scales could not be shown on (K, L, M, N, O, P, Q, R, S, T, U) (from patient 2) as original MRI data could not be obtained in a reasonable time frame because patient 2 passed away. (A, B, C, F, G, H, L, M, N, P, Q, R, S, T, U): T2-weighted images; (D, E, K, O): T1-weighted images; (I, J): fluid attenuated inversion recovery.

intermolecular hydrogen bonds with the main-chain oxygen atom of the $Na_V1.5$ Ile1892 and the $Na_V1.2$ Ile1896, respectively (Fig 6C and D). Replacing Glu87 with a lysine, which has a longer side chain, would disrupt this hydrogen bond and cause severe steric clashes with the $Na_V$ proteins. Thus, the *FGF12* Glu87Lys variant is likely to result in a significant loss of the ability to bind $Na_V$ proteins.

Table 1. Clinical features associated with *FGF12* aberrations.

| | This study | | Willemsen et al (2020) | | | Siekierska et al (2016) | | Takeguchi et al (2018) | |
|---|---|---|---|---|---|---|---|---|---|
| | Patient 1 | Patient 2 | Patient 3 | Patient 4 | Patient 5 (the mother of patient 4) | Patient 6 | Patient 7 (the younger brother of patient 6) | Patient 8 | Patient 9 |
| *FGF12* aberrations (GRCh38) | Biallelic SVs (Del/Dup) chr3: 192,083,678–192,145,703 (involving exon6, 3'UTR) chr3: 192,330,351–192,499,579 (involving exon 3–4) | NM_004113.6 c.259G>A p.(Glu87Lys) (Homozygous variant) | Heterozygous duplication chr3: 192,142,300–192,733,325 (involving exon1-6) | Heterozygous duplication chr3: 192,159,179–192,736,896 (involving exon1-5) | Heterozygous duplication chr3: 192,159,179–192,736,896 (involving exon1-5) | NM_004113.6 c.155G>A p.(Arg52His) (Heterozygous variant) | NM_004113.6 c.155G>A p.(Arg52His) (Heterozygous variant) | NM_004113.6 c.155G>A p.(Arg52His) (Heterozygous variant) | NM_004113.6 c.155G>A p.(Arg52His) (Heterozygous variant) |
| Sex | Female | Male | Male | Male | Female | Female | Male | Male | Male |
| Ethnicity | Japanese | Portuguese and African | N/A | N/A | N/A | Caucasian | Caucasian | Japanese | Japanese |
| Current age at reported or age at last examination | 8 yr 10 mo | Died at 11 yr because of COVID-19 pneumonia | 10 yr | 3 yr | 30 yr | Died at 7 yr because of status epilpticus | Died at 3 yr 6 mo (unknown cause) | 33 yr 3 mo | 2 yr 6 mo |
| Seizure type | Generalized tonic-clonic, focal motor, bilateral tonic-clonic, status epilepticus | Epileptic spasm, tonic, myoclonic jerks | Tonic-clonic, atonic, tonic, myoclonic, autonomic | Generalized tonic-clonic, history of febrile seizures, myoclonic jerks | Generalized tonic-clonic | Tonic, combined generalized and focal | Tonic, combined generalized and focal | Focal, tonic | Apnea attack, generalized tonic-clonic, focal seizure with pallor |
| Interictal EEG | Multifocal, multiple spike, hypsarrhythmia | Multiple discharges, hypsarrhythmia | Slow background, multifocal seizeures | Generalized epileptic activity | Normal | Slow background with multifocal epileptiform discharges, later hypsarrhythmia | Slow background with multifocal epileptiform discharges, later hypsarrhythmia | Suppression burst pattern, later hypsarrhythmia | Slow background with multifocal epileptiformm discharges |
| AED treatment (current or most recent) | Resistant to AEDs (Valproate, Zonisamide, Phenytoin, Phenobarbital) | Resistant to AEDs (Carbamazepine) | Resistant to AEDs | Valproate | Resistant to AEDs | Resistant to AEDs | Resistant to AEDs | Resistant to AEDs | Resistant to AEDs |

| | | This study | | Willemsen et al (2020) | | | Siekierska et al (2016) | | Takeguchi et al (2018) | |
|---|---|---|---|---|---|---|---|---|---|---|
| Seizure onset | Age at onset | 5 mo | 5–6 mo | 12 mo | 13 mo | 1 mo | 14 d | 1 mo | 7 d | 1 d |
| | Initial symptom | Spasm, developmental delay | Spasm, developmental delay | Seizures | Febrile convulsion | Seizures | Tonic seizures | Tonic seizures | Seizures | Apnea attack |
| Development | Head control | None | 8 mo | N/A | N/A | N/A | N/A | N/A | N/A | N/A |
| | Sitting | None | 11 mo | N/A | N/A | N/A | 24 mo | N/A | None | None at 1 yr 5 mo |
| | Walking | None | 3 yr | Wheelchair dependent | Walk independently (5 yr), uncoordinated gait | Walking with unsteadiness | None | N/A | None | N/A |
| | Meaningful words | Non verbal | Non verbal | No speech | Clearly speech, but vocabulary reduced | Speech problem | Non verbal | N/A | Non verbal | Non-verbal at 1 yr 5 mo |
| | Regression | 4 mo | No, always delayed but he seems to get worse when seizures are not under control | N/A | None | None | N/A | N/A | N/A | 5 mo |
| Degree of ID | | Severe | Moderate | Severe | Moderate | Mild to moderate | Severe | Severe | Severe | Severe |
| Movement disability | | No voluntary movement, myoclonus (hands) | Cerebellar ataxia with occasional dyskenetic movements, spasticity | Ataxia | Unsteady | Unsteady and migraine | Ataxia | Ataxia | Spastic dystonic quadriplegia, dystonic hypertonia of the neck and upper extremities | N/A |
| Behavioral features | | No smile, no eye contact | Autisti-like traits | None | Autism spectrum disorder | None | Stereo types, no eye contact | Stereo types, no eye contact | Stereo types, no eye contact | Poor eye contact |

**Table 1. Continued**

| | This study | Willemsen et al (2020) | Siekierska et al (2016) | Takeguchi et al (2018) |
|---|---|---|---|---|
| Brain MRI (CT) findings | Almost normal (7 mo), diffuse cerebral atrophy without obvious cerebeller atrophy (1 yr 7 mo); Normal myelination with a mild cerebral atrophy (12 mo and 1 yr 7 mo), gray matter atrophy with cortical atrophy (6 yr) | Mild prominence of the subarachnoid space in the frontal regions bilaterally otherwise normal; Bilateral delayed myelination in the parieto-occipital region | Normal (CT); Normal (5 mo), cerebeller atrophy (6 yr); Normal (2 mo), cerebeller atrophy (3 yr) | Normal (7 yr), mildly enlarged lateral ventricles (13 yr); Mild cerebral atorophy (6 mo), diffuse cerebral atrophy (1 yr 7 mo) |
| Other features | Cholecystitis, gallbladder removal at 8 yr; Hypotonia, tremors (early mo~2 yr), dysmorphic facial features (broad forehead, pointed chin) | Recurrent infection (upper airway), reflux, constipation, feeding difficulties; Dysmorphic facial features; Recurrent infections (perianal abscess, dental infections) | Acquired microcephaly, axial hypotonia, severe feeding difficulties, cerebral visual impairment; Acquired microcephaly, hypotonia, feeding difficulties | Microcephaly, multiple contractures of the extremities in flexion; Microcephaly |

Del: intragenic deletion allele, Dup: intragenic tandem duplication allele, AEDs: anti-epileptic drugs, CT: computed tomography, MRI: magnetic resonance imaging, N/A: no assesment.

Surprisingly, the previously reported recurrent SNV, Arg52His, exhibited a GOF effect on $Na_V$ in vitro cellular experiments (Siekierska et al, 2016), despite the fact that this variant is predicted to have weaker $Na_V$ protein binding. Although the side chain of Arg52 in the FGF12 complex is not visible (Fig 6C), its FGF13 equivalent (Arg57) forms a salt bridge and a hydrogen bond with Asp1856 and Glu1894 of $Na_V$1.2, respectively (Fig 6D). These structural consideration findings showed that substituting Arg52 with a histidine, which has a shorter side chain, would disrupt these interactions but may not cause severe steric clashes with adjacent residues, consistent with its retained ability to modulate $Na_V$ function.

### *Drosophila* in vivo functional analysis

To evaluate the functional significance of the *FGF12* p.Glu87Lys variant in vivo, we used a *Drosophila* system. *branchless* (*bnl*), an ortholog of *FGF12*, controls the branching morphogenesis of the *Drosophila* tracheal system through activation of the FGF signaling pathway by binding the FGF receptor Breathless (Btl) (Klämbt et al, 1992; Sutherland et al, 1996; Ghabrial et al, 2003). The *Drosophila* tracheal structure consists of dorsal longitudinal trunks that run from the anterior to posterior ends with regularly spaced branches (Uv et al, 2003). Previously, the ectopic expression of *bnl* in the epidermis showed an abnormal pattern of branching with masses of fine branches growing out from the stunted primary branches (Sutherland et al, 1996). We generated fly strains harboring UAS-Bnl or UAS-Bnl Glu324Lys (Glu87Lys in *FGF12*) at the attP40 site and mated them with the 69B-GAL4 driver. Of the 23 Bnl WT overexpressed embryos, 22 showed a dorsal trunk (DT) defect compared with no-treatment WT embryos, as previously reported (Fig S5A and B). However, Bnl Glu324Lys overexpressed embryos showed no defects in the tracheal system (N = 36) (Fig S5C), suggesting that *bnl* Glu324Lys is a loss-of-function variant.

## Discussion

We identified biallelic intragenic SVs and a homozygous SNV in *FGF12*, which have never been reported in association with epilepsy. Notably, biallelic intragenic SVs were overlooked by ES, but later detected by LRWGS in this study. Recently, biallelic CNV involvements have been revealed in Mendelian disorders (Yuan et al, 2020). Thus, LRWGS is highlighted as a powerful tool to detect disease-causing SVs in exome-negative individuals. We also performed OGM and successfully confirmed the presence of biallelic SVs in *FGF12*, but with some different sizes and breakpoints due to the different analytic principles between OGM and sequencing (Mantere et al, 2021) (Table S2).

In patient 1 with biallelic intragenic SVs in *FGF12*, ddPCR analysis confirmed the significantly diminished *FGF12* expression in LCL, suggesting complete LOF of *FGF12*. In vitro cellular experiments with the absence of FGF12 proteins, electrophysiological recordings of the $Na_V$1.6-derived sodium current have shown a LOF manner on $Na_V$-inactive gating compared with the presence of WT or GOF-mutant FGF12 proteins (Siekierska et al, 2016). This finding allows us

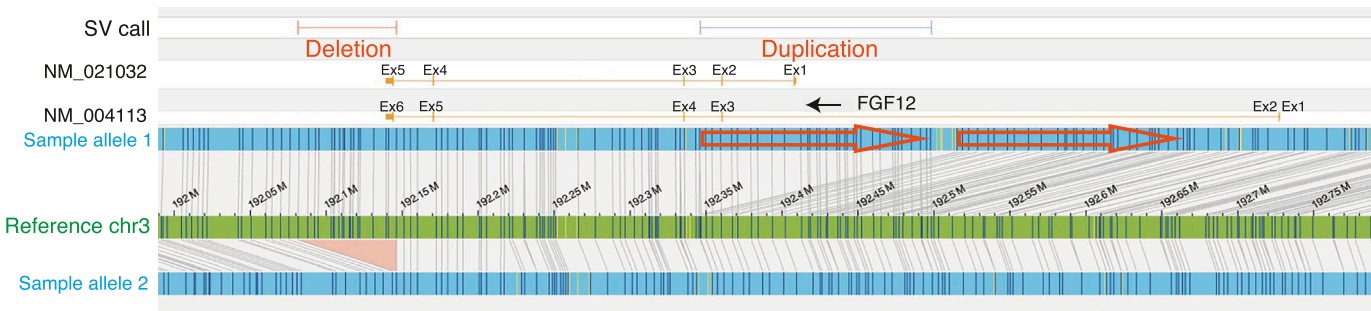

**Figure 3. Biallelic *FGF12* SVs analyzed by optical genome mapping.**
The reference genome (GRCh38) is represented by a green bar, and the alleles of patient 1 are shown in blue. Allele 1 has an intragenic duplication (shown by orange arrows), and allele 2 has a deletion (shown by an orange triangle). OGM with 102-fold effective genome coverage detected a 65-kb deletion (chr3:192,081,487–192,146,213) and a 152-kb duplication (chr3:192,346,230–192,498,352) including exon 3 in *FGF12*. *FGF12* is a reverse strand gene (a black arrow). Ex, exon.

to estimate how the complete LOF of *FGF12* would impair neuronal excitability by altering the function of Na$_V$s. Since Na$_V$1.2 and Na$_V$1.6, which interact with FGF12, are localized in the central nervous system and are expressed not only in excitatory but also in inhibitory neurons, impairing their neuronal excitability by the effect of FGF12 LOF on Na$_V$s may result in an imbalance of excitatory and inhibitory neuronal activities, eventually leading to epilepsy. Indeed, individuals with a LOF SNV in *SCN8A*/*SCN2A*, the genes encoding Na$_V$1.6/Na$_V$1.2, also exhibit epilepsy (Blanchard et al, 2015; He et al, 2019; Meisler, 2019; Johannesen et al, 2022), and the concept that impaired inhibitory neuronal excitability results in epilepsy has been considered in the related disorder of *FGF13*, a member of the FGF homologous factor family (Puranam et al, 2015).

In addition, ASE analysis of the healthy father of patient 1 by ddPCR suggested that the expression that was potentially compensated was observed from the normal allele but not from the deletion allele, showing an apparent allelic imbalance in the compensation of *FGF12* expression.

Transcripts from the aberrant allele with tandem duplication containing exons 3 and 4 were subjected to NMD. Heterozygous partial (exons 1–5, exons 1–6) duplication of *FGF12*, detected by CMA, has been reported in DEE individuals (patients 3–5) (Table 1) (Willemsen et al, 2020; Seiffert et al, 2022) and it has been suggested recently that transcripts of these aberrations harbor LOF effects on Na$_V$1.2/1.6 by functional analyses (Seiffert et al, 2022). However, the carrier mother in our current study was unaffected. These conflicting findings might be explained in part by differences in the exons and duplication sizes involved, or possibly by the missed detection of a hidden SV in the "normal allele." In addition, our gene expression analysis used patient LCLs, which are not of neuronal origin.

In patient 2, trio-based ES identified a homozygous variant [c.259G>A p.(Glu87Lys)] in *FGF12*. According to the structural consideration, p.Glu87Lys would cause severe steric clashes with Na$_V$ proteins, leading to loss of the ability to bind Na$_V$ proteins in accordance with the effects of FGF12 LOF on Na$_V$s. Another structural consideration was performed on the previously reported recurrent SNV in DEE individuals, [c.155G>A p.(Arg52His)], which exhibited a GOF effect on Na$_V$1.6 (Siekierska et al, 2016). This showed that substituting Arg52 with histidine retained the ability to modulate

Na$_V$ function. These data suggest complicated pathologies of *FGF12*-related disorders, in which various pathogenic variants affect molecular interactions in different ways, possibly resulting in complicated electrophysiological effects.

In addition, the heterozygous missense variant, [c.446C>A p.(Pro149Gln)] (rs17852067), which has never been reported in individuals with epilepsy or in gnomAD, was previously identified in individuals with arrhythmia (Li et al, 2017). According to the cellular experiments in vitro, this variant reduced the binding ability to Na$_V$1.2 and exhibited LOF effects on Na$_V$-inactive gating and decreased FGF12 localization to the axon initial segment in rat hippocampal neurons, which could lead to altered neuronal excitability (Wang et al, 2011). Despite the different locations of the two missense variants in *FGF12*, p.(Glu87Lys) in patient 2 and p.(Pro149Gln) likely lead to the loss of binding ability to Na$_V$s.

Our *Drosophila* in vivo functional analysis supported LOF of FGF12 with p.Glu87Lys. Bnl binds to the tracheal receptor tyrosine kinase Btl and activates the *bnl*/*btl* signaling pathway. The fact that Bnl p.Glu324Lys overexpression did not induce defects in the tracheal morphology suggests that Bnl p.Glu324Lys has lost the ability to activate the FGF receptor Btl. This view fits into a structural model in which p.Glu87Lys would lose the ability to bind Na$_V$ proteins (Fig 6C).

The phenotypic spectrum of the previously reported *FGF12*-related disorders includes tonic seizures, ID, speech problems, autistic features, and ataxia. Recently, LRS has revealed the causative genes of tremor-related neurodegenerative disorders, involving especially tandem repeat expansions or SVs (Marsili et al, 2022). Patient 2 manifested tremors, suggesting that the phenotypic spectrum of biallelic *FGF12*-related disorder may be potentially related to tremors. Biallelic LOF aberrations (in this study) and a heterozygous GOF missense variant of *FGF12* (patients 1, 2, and 6–9) had a more significant impact on the age of onset of seizures and the clinical severeness of DEE than heterozygous partial duplication of *FGF12* (patients 3–5) (Table 1) (Siekierska et al, 2016; Takeguchi et al, 2018; Trivisano et al, 2020; Willemsen et al, 2020). Brain MRI revealed severe diffuse cerebral atrophy with no cerebellar atrophy at 1 yr and 7 mo in patient 1 (Fig 2F–J and Table 1) and milder features (normal myelination with a mild cerebral atrophy) at 1 yr and 7 mo

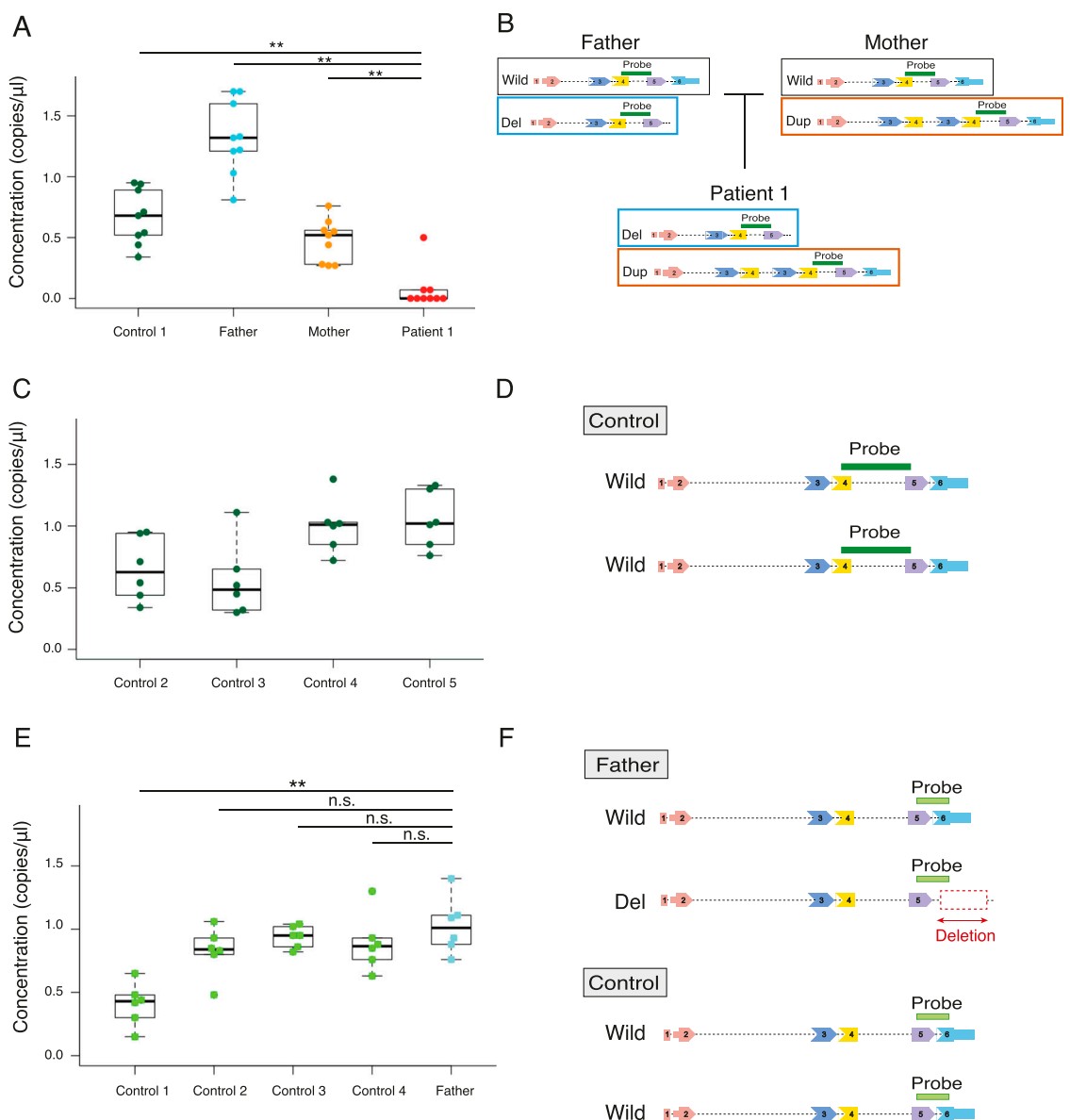

**Figure 4. Highly sensitive *FGF12* expression analysis in LCLs of family 1 and controls using ddPCR.**
**(A)** *FGF12* expression in LCLs of patient 1, her parents, and an unrelated control using ddPCR. The plot shows distributions of the absolute number of copies/μl of *FGF12* transcripts in LCLs (y = concentration [copies/μl]). Data are shown as box and whisker plots (25th–75th percentiles, median). Mann–Whitney *U* tests performed on medians of the triplicate reactions in three technical replicates (n = 9). **P < 0.01. **(B)** Schematic presentation of intragenic SVs in *FGF12* of family 1 and the position of the TaqMan probe used in the assay (A) (Table S3). Respective *FGF12* alleles in patient 1 and her parents are shown, with the location of the TaqMan probe covering exons 4 and 5. **(C)** *FGF12* transcript levels in LCLs of four unrelated controls by ddPCR. The plot shows the number of copies/μl of *FGF12* transcripts from the LCLs (y = concentration [copies/μl]). Data are shown as box and whisker plots (25th–75th percentiles, median). The ddPCR reactions were repeated twice in triplicates (n = 6). **(D)** Schematic presentation of the TaqMan probe position used in the assay (C) spanning exons 4 and 5 (Table S3). **(E)** The ASE analysis of *FGF12* transcript levels in LCLs of the father of patient 1 and unrelated controls using ddPCR. The plot shows distributions of the number of copies/μl of *FGF12* transcripts from the LCLs (y = concentration [copies/μl]). Data are shown as box and whisker plots (25th–75th percentiles, median). Mann–Whitney *U* test performed on medians of the triplicate reactions in two independent experiments (n = 6). **P < 0.01; n.s., no significance. **(F)** Schematic presentation of SV in the father and the position of the TaqMan probe used in the assay (E) (Table S3). As the TaqMan probe spanned exon 5 and the last exon (exon 6), the deletion allele could not be amplified in the father.
Source data are available for this figure.

in patient 2 (Fig 2O–R). This may be explained by possible different pathogenic impacts of *FGF12* aberrations (SV and missense variant) between the two patients.

In conclusion, we identified biallelic *FGF12* aberrations with LOF impacts on Na$_V$s in two DEE patients. Biallelic

intragenic SVs in *FGF12*, which were overlooked by ES, could successfully be found by LRWGS, promising us to find further patients with biallelic LOF SVs of *FGF12*. Furthermore, applying SV analysis using LRWGS after unsuccessful ES to DEE and neurological disorders can contribute to final diagnoses and

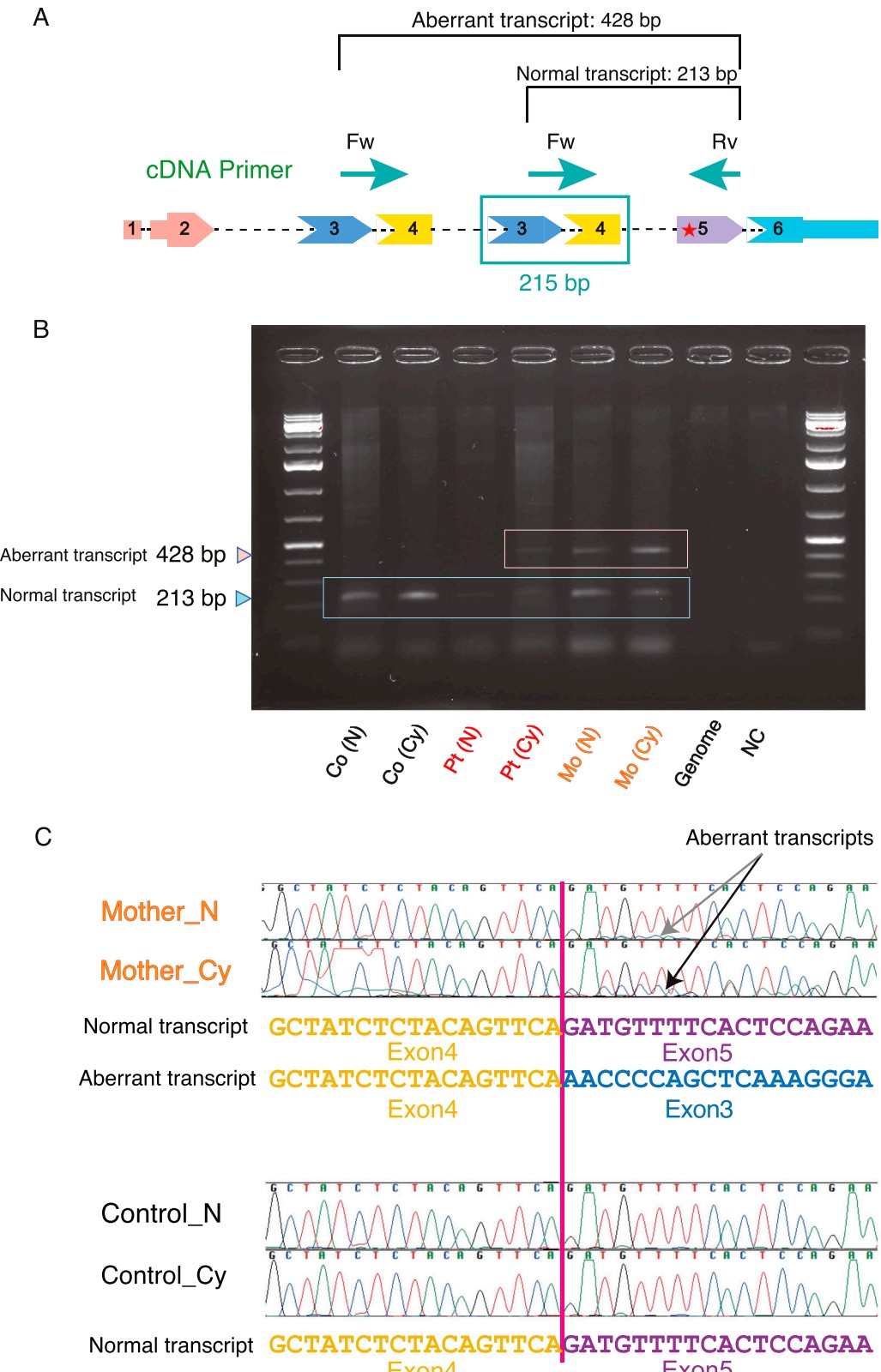

**Figure 5. Assessment of NMD of the tandem-duplication allele in patient 1 and her mother by Sanger sequencing.**
**(A)** Schematic presentation of a tandem duplication of exons 3 and 4 in *FGF12* of patient 1 and her mother. The predicted splicing pattern is shown by lines connecting the exons. As the duplicated segment included exons 3 and 4, of 215 bp in length, the normal and aberrant transcripts amplified by PCR were estimated to be 213 and 428 bases, respectively. A red asterisk in exon 5 indicates premature termination codons created by the duplication. The positions of primers used in this assay are drawn in

full understandings of genomic abnormalities in human diseases.

# Materials and Methods

### Samples and ethical issues

Genomic DNA of blood leukocytes from patients and their parents was examined after obtaining informed consent. The Institutional Review Board of the Yokohama City University Department of Medicine approved the experimental protocols for genetic analysis (the number, A19080001). Our research conformed to the principles of the Helsinki Declaration. Informed consent was also obtained from family 2 to publish photos and a movie of patient 2.

### Controls in gene expression analyses

Total RNA was extracted from LCLs to confirm *FGF12* expression in patient 1 and her parents, and three healthy controls and four controls with diseases other than epileptic phenotypes (Table S4). All exome data of disease controls were confirmed with no pathogenic SNVs or copy aberrations in *FGF12*.

### Trio-based ES

For patients and their parents in this study, library preparation, sequencing, data acquisition, processing, variant calling, annotation, and filtering for rare variants were carried out as previously described (Seyama et al, 2022). SNVs were evaluated using SIFT (Kumar et al, 2009), PolyPhen-2 (Adzhubei et al, 2013), and CADD (Rentzsch et al, 2019).

### Detecting CNVs from exome data

We tried to detect candidate pathogenic CNVs using exome data with XHMM (Fromer et al, 2012; Miyatake et al, 2015) and, if necessary, a method developed by Nord et al with some modifications (jNord) as previously described (Nord et al, 2011; Uchiyama et al, 2021).

### LRS using PromethION

Genomic DNA was extracted from the blood leukocytes of patient 1 using the standard method. The library was prepared for nanopore sequencing using a DNA ligation kit (SQK-LSK109) and applied to a PromethION sequencer using one PRO-002 (R9.4.1) flow cell (all Oxford Nanopore Technologies). Base calling and fastq conversion were carried out with MinKNOW version 20.06.9. We also sequenced

control datasets by PromethION as previously described (Mitsuhashi et al, 2020).

### Detecting SVs from LRWGS

We aligned nanopore long reads to the human reference genome (GRCh38) using LAST. The analytic process for detecting and characterizing SVs is overviewed as previously described (Mitsuhashi et al, 2020). Concisely, dnarrange was used to find SVs, and patient-specific SVs were selected using 30 control datasets. Then, using lamassemble, each group of overlapping SV reads was merged into a consensus sequence and realigned to the reference genome.

### Manual assessment of patient 1-specific SVs and gene annotation

To rule out benign SVs from patient 1, we assessed patient-specific SVs using the Database of Genomic Variants (MacDonald et al, 2014), which comprises genomic SVs observed in healthy individuals. We selected all disrupted, duplicated, and deleted protein-coding genes, considering the gene information from OMIM and HGMD Professional, along with their respective pLI scores (Lek et al, 2016).

### OGM

OGM was performed for patient 1, harboring biallelic SVs. Ultra-high molecular weight DNA was extracted from 1.5 million lymphoblastoid cells using the SP Blood and Cell Culture DNA Isolation Kit according to the manufacturer's instructions (Bionano Genomics) (Mantere et al, 2021; Bruijn et al, 2022). DNA was labeled using a DLS Labeling Kit (Bionano Genomics). The labeled DNA was loaded on a Saphyr chip G2.3 and run on a Saphyr instrument (Bionano Genomics) for an output of 400 Gb. De novo assembly and variant annotation of the OGM data were conducted on Bionano Solve software version 3.7. We focused on *FGF12* and visualized it on Bionano Access 1.7.1.1.

### Segregation analysis and confirmation of SV breakpoints and SNVs by Sanger sequencing

PCR primers for the SV breakpoints and SNVs were designed using primer3 software and are listed in Table S3. PCR products using Ex Taq and LA Taq (Takara Bio Inc.) were sequenced using BigDye Terminator v3.1 Cycle Sequencing kit with 3130xL/3500xL genetic analyzers (Applied Biosystems).

---

green arrows. cDNA, complementary DNA; Fw, forward primer; Rv, reverse primer. **(B)** Gel electrophoresis of PCR products amplifying normal or aberrant transcripts using cDNA derived from LCLs of patient 1 and her mother. The smaller (213 bp) and larger (428 bp) bands indicate normal and aberrant transcripts, respectively. Co, control; Pt, patient 1; Mo, mother of patient 1; N, treated without CHX (no treatment); Cy, treated with CHX; NC, negative control (no cDNA). **(C)** Confirmation of NMD in cDNA from a tandem duplication allele. The cDNA sequences of the PCR products showed that cDNA derived from the tandem duplication allele from the mother of patient 1 treated with CHX had higher peaks (black arrow) than that without CHX treatment (gray arrow). No such changes were detected in a control. The primers used in this assay are listed in Table S3 and described in Fig 5A.
Source data are available for this figure.

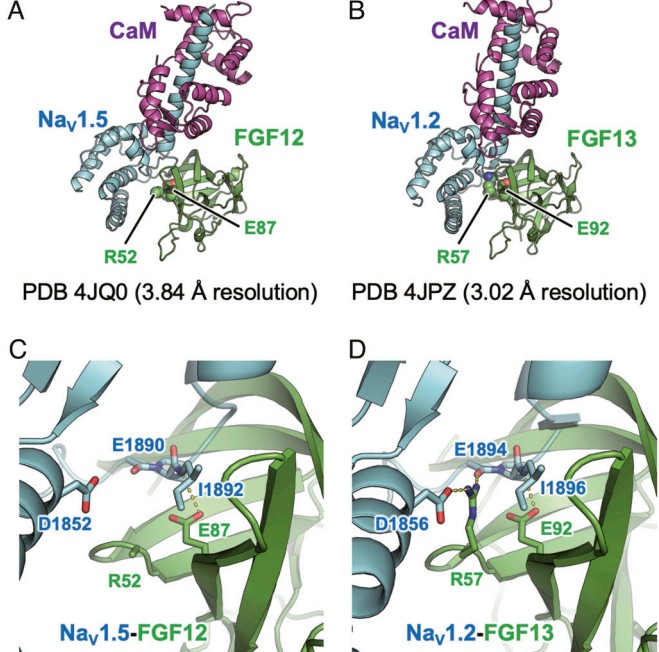

**Figure 6. Structural considerations of FGF12-Na$_V$1.5 and FGF13-Na$_V$1.2.**
**(A, B)** FGF12 Glu87 in (A) and its equivalent residue (FGF13 Glu92 in (B)) are located at the interface with Na$_V$ proteins. CaM, calmodulin. **(C, D)** FGF12 Glu87 in (C) and FGF13 Glu92 in (D) form intermolecular hydrogen bonds with the main-chain oxygen atom of the Na$_V$1.5 Ile1892 and the Na$_V$1.2 Ile1896, respectively. The side chain of Arg52 in the FGF12 complex is not visible; however, its FGF13 equivalent (Arg57) forms a salt bridge and a hydrogen bond with Asp1856 and Glu1894 of Na$_V$1.2.

### ASE analysis of *FGF12* mRNA using LCL by ddPCR

We performed ASE analysis of *FGF12* mRNA using LCL of patient 1 and her parents by ddPCR with high sensitivity. LCLs derived from patient 1, her parents, and unrelated control individuals were treated with or without (i.e., no treatment) 0.03% (vol/vol) of DMSO, as a vehicle control, and with 30 μg/ml of CHX, an NMD inhibitor, at 37°C for 6 h. Total RNA was extracted from LCLs using an RNeasy Plus Mini Kit (QIAGEN), and cDNA was synthesized by reverse transcription PCR (RT–PCR) from 5 μg of total RNA with SuperScript III (Thermo Fisher Scientific). The ddPCR and data analysis using a Droplet Digital PCR QX200 system (Bio-Rad Laboratories) and QuantaSoft (version 1.7.4.0917; Bio-Rad Laboratories) were used as described previously (Tsuchida et al, 2021). The ddPCR reaction mixtures (20 μl) contained 10 μl of ddPCR Supermix for probes (no dUTP, 2×; Bio-Rad Laboratories), 1.0 μl of TaqMan probe, and 100 ng of template cDNA (no treatment) for one reaction. The thermal cycling conditions were as follows: 95°C for 10 min; 40 cycles of 94°C for 30 s, and 60°C for 120 s; and 98°C for 10 min. The ddPCR assays were repeated twice or three times in triplicate. The TaqMan probes used in the assay are listed in Table S3 and described in Fig 4B, D, and F.

### Statistical analysis

Box and whisker plots depict the median and ranges from the first to the third quartile. We used EZR software (Kanda, 2013) for

statistical analysis. *FGF12* expression level in controls, patient 1, and her parents were compared using a Mann–Whitney *U* test. A *P*-value less than 0.01 was set as the threshold for statistical significance.

### Assessment of NMD of the maternal aberrant allele with tandem duplication by electropherogram peak heights of Sanger sequencing

In general, up-regulation of aberrantly spliced transcripts by inhibition of NMD provides supportive evidence that premature termination codon-containing mRNAs are under the control of NMD (Asselta et al, 2010). In addition, Sanger sequencing electropherogram peak heights are highly dependent on the amount of cDNA template. Therefore, qualitative analysis of alterations in the signal peak heights of Sanger sequencing with inhibition of NMD by CHX treatment, compared with no treatment, allows us to estimate whether transcripts from the aberrant allele are subjected to NMD.

The cDNA synthesis and Sanger sequencing procedures were carried out as described above (Methods). The PCR reaction mixtures with LA Taq contained 240 ng of template cDNA for one reaction. The thermal cycling conditions were as follows: 94°C for 1 min; 35 cycles of 94°C for 30 s, 60°C for 30 s, and 72°C for 2 min; and 72°C for 5 min. The primers used in this assay (the primer positions were forward, spanning exons 3 and 4; and reverse, within exon 5) are listed in Table S3 and described in Fig 5A.

### Structural considerations

The coordinates of the crystal structures were obtained from the Protein Data Bank (ID 4JQ0 and 4JPZ). Structural analyses were conducted with the program PyMOL (Schrödinger, LLC).

### Generation of transgenic fly strains

Bnl cDNA was chemically synthesized according to FBtr0014135 and introduced into the pEX-A2J2 vector (Eurofins). The resulting pEX-A2J2-Bnl was amplified in *Eschericia Coli*. Bnl cDNA was excised by a restriction enzyme and transferred into the pBFv-UAS3 vector (#138399; Addgene) to generate pBFv-UAS3-Bnl. To generate pBFv-UAS3-Bnl (Glu324Lys), pBFv-UAS3-Bnl was used as the template for PCR-based mutagenesis with the QuikChange Site-Directed Mutagenesis kit (Agilent) and the specific primers shown in Table S3. The resulting vectors were integrated into the attP40 landing site as described previously (Hamanaka et al, 2022). The UAS3-Bnl and UAS3-Bnl (Glu324Lys) lines used in this study have the *y cho v*; attp40{UAS-Bnl or Bnl (Glu324Lys)} genotype. 69B-GAL4 (#106–499; DGRC) flies were obtained from the KYOTO Stock Center at Kyoto Institute of Technology.

### Immunohistochemistry of whole-mount embryos

Immunohistochemistry was basically performed as previously described (Araújo & Aslam, 2005). Embryos were fixed with 4% paraformaldehyde in PBS:heptane (1:4 ratio) for 20 min. The embryos were then devitellinized in methanol:heptane (5:4 ratio).

Embryos were blocked by PBSBT (0.2% Triton X-100, 0.2% Tween20, and 3 drops horse serum in 10 ml PBS) and were incubated with the anti-GASP antisera (2A12: #AB528492; DSHB) as the primary antibody. After washing with PBST, the avidin–biotin complex (ABC) method was applied to embryos for amplification of the signal intensity (VECTASTAIN Elite ABC Kit Peroxidase [Mouse IgG]; DAB Substrate Kit, Vector Laboratories). After the ABC reaction, embryos were mounted on slides with VECTAMount Permanent Mounting Medium (Vector Laboratories). Images were taken by an inverted microscope (IX73; Olympus) and the cellSens Standard software (Olympus) and were processed by Photoshop software (Adobe Inc).

## For more information

### Web resources
gnomAD (Genome Aggregation Database): https://gnomad.broadinstitute.org

PolyPhen-2: http://genetics.bwh.harvard.edu/pph2/

SIFT: https://sift.bii.a-star.edu.sg/

CADD: https://cadd.gs.washington.edu/

LAST: https://github.com/mcfrith/last-rna/blob/master/last-long-reads.md.

dnarrange: https://github.com/mcfrith/dnarrange.

lamassemble: https://gitlab.com/mcfrith/lamassemble

Database of genomic variants: http://dgv.tcag.ca/dgv/app/home

OMIM (Online Mendelian Inheritance in Man): https://omim.org/

Human Gene Mutation Database (HGMD) Professional: https://portal.biobase-international.com/hgmd/pro/start.php

primer3: http://bioinfo.ut.ee/primer3-0.4.0/

GTEx portal: https://gtexportal.org/home/

The MANE project: https://www.ncbi.nlm.nih.gov/refseq/MANE/

UCSC genome browser: https://genome.ucsc.edu/.

# Data Availability

This study includes no data deposited in external repositories.

# Supplementary Information

# Acknowledgements

We thank all patients and family members for participating in this study. We also thank K Takabe, T Miyama, M Sato, S Sugimoto, and N Watanabe for technical assistance and the *Drosophila* Genetic Resource Center, Kyoto Institute of Technology for providing the fly strains. This work was supported by Japan Agency for Medical Research and Development (AMED) under grant numbers JP22ek0109484 (K Saito), JP22ek0109486, JP22ek0109549, and JP22ek0109493 (N Matsumoto); JSPS KAKENHI under grant numbers JP20K17936 and JP22K15901 (A Fujita), JP20K07907 (S Miyatake), JP20K08164 (T Mizuguchi), JP20K16932 (K Hamanaka), JP21K07869 (E Koshimizu), JP21K15907 (Y Uchiyama), and JP20K17428 (N Tsuchida); and the Takeda Science Foundation (T Mizuguchi and N Matsumoto). We thank Catherine Perfect, MA (Cantab), from Edanz (https://jp.edanz.com/ac), for editing a draft of this manuscript.

## Author Contributions

S Ohori: conceptualization, data curation, formal analysis, validation, investigation, visualization, methodology, and writing—original draft, review, and editing.

A Miyauchi: resources, and writing—original draft, review, and editing.

H Osaka: resources, and writing—original draft, review, and editing.

CM Lourenco: resources, and writing—original draft, review, and editing.

N Arakaki: data curation, investigation, visualization, methodology, and writing—original draft, review, and editing.

T Sengoku: data curation, investigation, visualization, methodology, and writing—original draft, review, and editing.

K Ogata: data curation, investigation, visualization, methodology, and writing—original draft, review, and editing.

RS Honjo: resources and writing—review and editing.

CA Kim: resources, data curation, investigation, visualization, methodology, and writing—original draft, review, and editing.

S Mitsuhashi: conceptualization, software, investigation, methodology, and writing—review and editing.

MC Frith: resources, software, methodology, and writing—review and editing.

R Seyama: resources, investigation, methodology, and writing—review and editing.

N Tsuchida: funding acquisition, methodology, and writing—review and editing.

Y Uchiyama: resources, software, funding acquisition, methodology, and writing—review and editing.

E Koshimizu: funding acquisition, methodology, and writing—review and editing.

K Hamanaka: conceptualization, software, funding acquisition, investigation, methodology, and writing—review and editing.

K Misawa: resources, software, methodology, and writing—review and editing.

S Miyatake: funding acquisition, methodology, and writing—review and editing.

T Mizuguchi: software, funding acquisition, methodology, and writing—review and editing.

K Saito: data curation, funding acquisition, investigation, visualization, methodology, and writing—original draft, review, and editing.

A Fujita: conceptualization, data curation, formal analysis, funding acquisition, validation, investigation, visualization, methodology, and writing—original draft, review, and editing.

N Matsumoto: conceptualization, supervision, funding acquisition, methodology, project administration, and writing—original draft, review, and editing.

## Conflict of Interest Statement

The authors declare that they have no conflict of interest.

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
