## [Reviewer comments · Life Science Alliance]

Life Science Alliance

Biallelic structural variations within FGF12 detected by long-read sequencing in epilepsy

Sachiko Ohori, Akihiko Miyauchi, Hitoshi Osaka, Charles Lourenco, Naohiro Arakaki, Toru Sengoku, Kazuhiro Ogata, Rachel Honjo, Chong Kim, Satomi Mitsuhashi, Martin Frith, Rie Seyama, Naomi Tsuchida, Yuri Uchiyama, Eriko Koshimizu, Kohei Hamanaka, Kazuharu Misawa, Satoko Miyatake, Takeshi Mizuguchi, Kuniaki Saito, Atsushi Fujita, and Naomichi Matsumoto
DOI: <https://doi.org/10.26508/lsa.202302025>

Corresponding author(s): Naomichi Matsumoto, Yokohama City University Graduate School of Medicine and Sachiko Ohori, Yokohama City University

Review Timeline:

Submission Date:	2023-03-06
Editorial Decision:	2023-03-29
Revision Received:	2023-05-10
Editorial Decision:	2023-05-11
Revision Received:	2023-05-22
Accepted:	2023-05-23

Scientific Editor: Novella Guidi

Transaction Report:

March 29, 2023

Re: Life Science Alliance manuscript #LSA-2023-02025-T

Naomichi Matsumoto
Yokohama City University Graduate School of Medicine
Yokohama

Dear Dr. Matsumoto,

Thank you for submitting your manuscript entitled "Biallelic structural variations within FGF12 detected by long-read sequencing in epilepsy" to Life Science Alliance. The manuscript was assessed by expert reviewers, whose comments are appended to this letter. We invite you to submit a revised manuscript addressing the Reviewer comments.

Thank you for this interesting contribution to Life Science Alliance. We are looking forward to receiving your revised manuscript.

Sincerely,

B. MANUSCRIPT ORGANIZATION AND FORMATTING:

Reviewer #1 (Comments to the Authors (Required)):

The authors here describe the discovery of two biallelic intragenic structural variations (SVs) in FGF12 gene identified by applying long-read whole genome sequencing (LRWGS) on two young patients with epileptic encephalopathy (DEE). To validate the molecular pathomechanisms of these biallelic FGF12 SVs/SNV, highly sensitive gene expression analyses using lymphoblastoid cells from the patient with biallelic SVs, structural considerations, and *Drosophila* in vivo functional analysis of the SNV were performed, confirming the loss-of-function. They conclude that LRWGS is important in identifying these mutations. The manuscript is of interest and needs only some improvement, mainly on the following aspects:

- I am curious about the presence of tremor/tremulousness in your patient(s). A recent systematic review has documented the presence of tremors in about 45% of cases studied using long-read sequencing (LRS) in neurodegenerative disorder. I was wondering if the authors could argue on these aspects, perhaps in the "The paper explained" section (See also Marsili L, Duque KR, Bode RL, Kauffman MA, Espay AJ. Uncovering Essential Tremor Genetics: The Promise of Long-Read Sequencing. *Front Neurol.* 2022 Mar 23;13:821189. doi: 10.3389/fneur.2022.821189).
- The authors mention about long-read whole genome sequencing (LRWGS) in the introduction, but what about LRS without WGS? Please, argue.
- I would expand the conclusions section a bit, better highlighting the next-steps related to these observations. Shall we use more frequently LRWGS in developmental epilepsy disorders (perhaps with clear family history)?

Minor comments;

- When describing the treatments with anti-epileptic drugs (AEDs) in the two patients, please add dosages of each AED.
- Page 7, Line 142: What does it mean "sometimes increased with fever"? Please, be more specific.
- Page 8, Line 162: " and at 6 years " please, add "of age".
- Page 10, Line 217: "Of the father" Please, add "of Patient 1"
- Page 23, Line 497: I believe it is "Escherichia Coli".

Reviewer #2 (Comments to the Authors (Required)):

This study reports for the first time biallelic intragenic SVs and a homozygous SNV in FGF12 in two patients with epilepsy by applying long-read whole genome sequencing. Expression analysis and in silico structural predictions of the interaction between FGF12 and NaV were done to support that epilepsy is caused by loss-of-function (LoF) due to biallelic FGF12 alterations. Moreover, in vivo functional analyses in *Drosophila* were performed.

The study is well conducted and reports novel findings, as heterozygous FGF12 mutations had previously been reported in DEE but never as biallelic mutations. This study further suggests a mechanism by LoF.

Comments

- Overall, the manuscript and figures should be shortened.
- The meaningful contribution on the part of *Drosophila* in vivo functional analysis is not very convincing and indirectly suggests a LoF of the missense mutation. It seems not to be essential to the manuscript. I, therefore, suggest removing it.
- Authors should further discuss to which extent the phenotype of patients with homozygous FGF12 mutations is more severe than those with heterozygous.

*Comments from the reviewer(s):**Reviewer #1 (Comments to the Author):*

The authors here describe the discovery of two biallelic intragenic structural variations (SVs) in FGF12 gene identified by applying long-read whole genome sequencing (LRWGS) on two young patients with epileptic encephalopathy (DEE). To validate the molecular pathomechanisms of these biallelic FGF12 SVs/SNV, highly sensitive gene expression analyses using lymphoblastoid cells from the patient with biallelic SVs, structural considerations, and Drosophila in vivo functional analysis of the SNV were performed, confirming the loss-of-function. They conclude that LRWGS is important in identifying these mutations.

The manuscript is of interest and needs only some improvement, mainly on the following aspects:

I am curious about the presence of tremor/tremulousness in your patient(s). A recent systematic review has documented the presence of tremors in about 45% of cases studied using long-read sequencing (LRS) in neurodegenerative disorder. I was wondering if the authors could argue on these aspects, perhaps in the "The paper explained" section (See also Marsili L, Duque KR, Bode RL, Kauffman MA, Espay AJ. Uncovering Essential Tremor Genetics: The Promise of Long-Read Sequencing. Front Neurol. 2022 Mar 23;13:821189. doi: 10.3389/fneur.2022.821189).

> Thank you very much for kindly evaluating our manuscript and providing us insightful comments.

Patient 2 manifested tremors, we therefore described them in the clinical features section, specifically on page 8, lines 167, 176, and Table 1. Patient 1 did not have tremors.

Based on the other reviewer comment, "The Paper Explained" section was removed to shorten the revised manuscript. Therefore, we instead added the above description in discussion section: page 17, lines 381-385. We hope this change is satisfactory to you.

The authors mention about long-read whole genome sequencing (LRWGS) in the introduction, but what about LRS without WGS? Please, argue.

> We mentioned "LRWGS" and "target-LRS" as well in the introduction section, page 5 lines 93-97.

- I would expand the conclusions section a bit, better highlighting the next-steps related to these observations. Shall we use more frequently LRWGS in developmental epilepsy disorders (perhaps with clear family history)?

> We have expounded upon our expectation toward LRS in the concluding section, specifically on page 18, lines 401-404.

Minor comments;

- When describing the treatments with anti-epileptic drugs (AEDs) in the two patients, please add dosages of each AED.

> We have added them.

- Page 7, Line 142: What does it mean "sometimes increased with fever"? Please, be more specific.

> Her seizures got more frequent during febrile episodes. (specifically on page 8, lines 156.)

- Page 8, Line 162: " and at 6 years " please, add "of age".

> Added (page 9, line 180).

- Page 10, Line 217: "Of the father" Please, add "of Patient 1"

> Added (page 11, lines 236).

- Page 23, Line 497: I believe it is "Escherichia Coli".

> Changed (page 24, line 526).

Reviewer #2 (Comments to the Author):

This study reports for the first time biallelic intragenic SVs and a homozygous SNV in FGF12 in two patients with epilepsy by applying long-read whole genome sequencing. Expression analysis and in silico structural predictions of the interaction between FGF12 and NaV were done to support that epilepsy is caused by loss-of-function (LoF) due to biallelic FGF12 alterations. Moreover, in vivo functional analyses in Drosophila were performed.

The study is well conducted and reports novel findings, as heterozygous FGF12 mutations had previously been reported in DEE but never as biallelic mutations. This study further suggests a mechanism by LoF.

> Thank you very much for your favorable comments.

Comments

- Overall, the manuscript and figures should be shortened.

> We tried to shorten our revised manuscript: Figures 4-6 were merged to new Fig. 4. The initial part of the discussion and the section of 'The Paper Explained' were omitted to avoid the redundancy, and was removed.

- The meaningful contribution on the part of Drosophila in vivo functional analysis is not very convincing and indirectly suggests a LoF of the missense mutation. It seems not to be essential to the manuscript. I, therefore, suggest removing it.

> Thank you very much for your suggestion. As we believe it is still important to show the loss of function aspect by missense variants, we retain the Drosophila functional data in the revised manuscript but move it to the supplemental materials (Fig S5). We hope this change is acceptable.

- Authors should further discuss to which extent the phenotype of patients with homozygous FGF12 mutations is more severe than those with heterozygous.

> The clinical manifestations of patients with biallelic structural variations (SVs) or missense variant were further discussed based on the symptoms exhibited by previously reported patients carrying heterozygous aberrations in discussion section (page 17, lines 385-388).

May 11, 2023

RE: Life Science Alliance Manuscript #LSA-2023-02025-TR

Naomichi Matsumoto
Yokohama City University Graduate School of Medicine
Yokohama

Dear Dr. Matsumoto,

Thank you for submitting your revised manuscript entitled "Biallelic structural variations within FGF12 detected by long-read sequencing in epilepsy". We would be happy to publish your paper in Life Science Alliance pending final revisions necessary to meet our formatting guidelines.

- please upload the main and supplementary figures as single files
- please upload your video file as a single file that is not in a zip file format
- please add ORCID ID for both corresponding authors-you should have received instructions on how to do so

Figure Check:

- Figure 2 and Figure S5 need a scale bar
- Fig. 5B and Figure S1B first column looks like a possible duplicate. Please provide source data for both panels.

A. FINAL FILES:

B. MANUSCRIPT ORGANIZATION AND FORMATTING:

Sincerely,

Reviewer #1 (Comments to the Authors (Required)):

The authors have successfully addressed all my points. I do not have any further comments. Good job!

Thank you very much for your constructive comments.

With regards to Figure 2, we could put scales only in patient 1 but not in patient 2 due to unavailability of original MRI data within 7days in patient 2. We also added a comment in the legend why we could not put scales of MRI in patient.

We also uploaded the main and supplementary figures as separated single files.

The images of the gels presented in Figure 5B and Figure S1B are actually different, representing RT-PCR and genomic PCR, respectively. We uploaded source data of both figures.

May 23, 2023

RE: Life Science Alliance Manuscript #LSA-2023-02025-TRR

Naomichi Matsumoto
Yokohama City University Graduate School of Medicine
3-9 Fukuura, Kanazawa-ku
Yokohama 236-0004
Japan

Dear Dr. Matsumoto,

Thank you for submitting your Research Article entitled "Biallelic structural variations within FGF12 detected by long-read sequencing in epilepsy". It is a pleasure to let you know that your manuscript is now accepted for publication in Life Science Alliance. Congratulations on this interesting work.

DISTRIBUTION OF MATERIALS:

Again, congratulations on a very nice paper. I hope you found the review process to be constructive and are pleased with how the manuscript was handled editorially. We look forward to future exciting submissions from your lab.

Sincerely,
